# Low-dimensional heat conduction in surface phonon polariton waveguide

Yu Pei [1,4], Li Chen[2,3,4], Wonjae Jeon[1,4], Zhaowei Liu [2,3] ✉ & Renkun Chen [1,3] ✉

Heat conduction in solids is typically governed by the Fourier's law describing a diffusion process due to the short wavelength and mean free path for phonons and electrons. Surface phonon polaritons couple thermal photons and optical phonons at the surface of polar dielectrics, possessing much longer wavelength and propagation length, representing an excellent candidate to support extraordinary heat transfer. Here, we realize clear observation of thermal conductivity mediated by surface phonon polaritons in $SiO_2$ nanoribbon waveguides of 20-50 nm thick and 1-10 μm wide and also show non-Fourier behavior in over 50-100 μm distance at room and high temperature. This is enabled by rational design of the waveguide to control the mode size of the surface phonon polaritons and its efficient coupling to thermal reservoirs. Our work laid the foundation for manipulating heat conduction beyond the traditional limit via surface phonon polaritons waves in solids.

In diffusive heat conduction in solids, thermal conductance is proportional to the cross-section area of the solid and inversely proportional to its length under the traditional Fourier heat transfer systems. Non-Fourier heat conduction has only been observed in exotic systems under unusual conditions, such as phononic and electronic waveguides at sub-Kelvin temperature[1,2], atomic and molecular structures[3–5], carbon nanotubes[6], and graphene[7]. Surface phonon polariton (SPhP) in polar dielectric solids possess long wavelength, on the order of a few μm, and thus could lead to non-Fourier thermal conduction in solid structures with larger dimensions around room temperature. SPhP are evanescent surface waves due to the collective oscillation of atoms on the surface of polar dielectric materials induced by electromagnetic (EM) waves, i.e., coupling between optical phonons and photons. The propagating length of SPhP can be much longer than the mean free path (MFP) of phonons at room and high temperatures, greater than 1 mm when the film thickness is less than 100 nm[7–9]. In principle, these propagating surface waves can contribute to heat conduction of the solid, especially in nanostructures where the volumetric heat conduction by phonons and electrons is relatively smaller. Unlike thermal radiation by photons, SPhP waves can carry high energy flux along the solid surface, i.e., the solid acts as a

thermal waveguide, akin to an optical fiber carrying light signals. Despite the early theoretical predictions[8,9] and their pronounced role in radiation heat transfer[10–12], observing heat conduction by SPhP in solids has been elusive due to its relatively small contribution to thermal conductance compared to classical heat carriers such as phonons and electrons. In fact, this new mechanism of heat conduction mediated by SPhP has been suggested in a few recent experiments[13,14]. However, the experimental work has not observed an increase in absolute thermal conductivity compared to the classical phonon heat conduction of the corresponding solids, but rather relied on the temperature dependence of the total thermal conductivity, which remains lower than that of the bulk phonon thermal conductivity. The uncharted regime of polaritonic heat conduction is exciting but has not been unambiguously realized in prior work.

To clearly differentiate the SPhP contribution from phonons, the cross-section of the waveguide has to be considerably reduced. This leads to greatly enhanced SPhP propagation length, which is highly desirable, but also drastically enlarged SPhP mode size which introduces two additional major concerns. First, the waveguide must be suspended far away from any bulky solids to avoid significant SPhP wave leakage. Second, the large mode size usually makes the coupling

[1]Department of Mechanical and Aerospace Engineering, University of California San Diego, 9500 Gilman Drive, MC 0411, La Jolla, CA 92093, USA. [2]Department of Electrical and Computer Engineering, University of California San Diego, 9500 Gilman Drive, MC 0407, La Jolla, CA 92093, USA. [3]Program in Materials Science and Engineering, University of California San Diego, 9500 Gilman Drive, MC 0418, La Jolla, CA 92093, USA. [4]These authors contributed equally: Yu Pei, Li Chen, Wonjae Jeon. ✉e-mail: zhaowei@ucsd.edu; rkchen@ucsd.edu

between the miniature waveguide and the thermal reservoirs much less efficient.

We realize that all the prior modeling and experiment works were only focused on the waveguide itself, such as the materials and geometries of nanowires and thin films, without considering the interaction of SPhP with the surrounding environment and reservoirs. However, to experimentally observe the polaritonic heat conduction, the aforementioned two concerns are crucial but have yet to be carefully addressed. In this work, through the rational design of the waveguide cross-section and its coupling to the reservoirs using an efficient absorber, we observe an increased thermal conductivity in nanoribbons (NRs) of $SiO_2$, by as much as 34% over its well-known phonon thermal conductivity limit. For samples with relatively small mode size which means leakage of the wave to the substrate during propagation is negligible, the thermal transport is shown to be ballistic over 100 μm length, that is, near constant thermal conductance with different lengths. Our results unambiguously demonstrate an enhanced effective thermal conductivity due to SPhP and the extraordinary characteristics compared to the traditional heat conduction carried by phonons and electrons, thus opening an entirely new regime for thermal transport.

## Results

Figure 1a shows our design for observing the polaritonic heat conduction in a $SiO_2$ NR waveguide. The NRs have thickness ($t$), width ($W$), and length ($L$) in the range of 20-50 nm, 1–10 μm, and 50–400 μm, respectively. Figure 1b shows the scanning electron microscopy images of the devices. The device fabrication process and additional images of the measured samples are shown in Supplementary Notes 1 and 2. The device consists of a heating and a sensing beam, representing the hot and cold thermal reservoirs, and the NR waveguide bridging the two. The $SiO_2$ in the beams and layers are fabricated from the same 100 nm-thick thermal oxide layer on a Si wafer and thus have

monolithic junctions without thermal contact resistance. The final NR thickness is reduced to 13– 44 nm via controlled etching using buffered HF etchant and the slow parasitic etching by $XeF_2$ (Supplementary Notes 1 and 3). The beams are covered by ~110 nm thick Pt used for joule heating and thermometry. The beams and the NRs are suspended ~ 70 μm away from the substrate to avoid major SPhP leakage and to enable high-resolution heat conduction measurement (~10 pW K⁻¹) using a modulated heating technique and a noise-canceling bridge scheme, as demonstrated in our earlier work[15]. The schematic of the measurement platform and the principle of the thermal conductivity measurement are in Supplementary Notes 4 and 5. The thickness of the NRs after etching was determined from the frequency-dependent thermal transport measurements at room or low temperatures that yielded heat capacity of the NR[16,17], which was used to determine the exact cross-section area of the NRs (Supplementary Note 6). This approach was further validated by atomic force microscopy (AFM) thickness measurement of selective NRs cut and transferred to a flat substrate (Supplementary Fig. S7).

### Design of $SiO_2$ nanoribbons waveguide

We designed the NRs and their coupling to the reservoirs by theoretically investigating the propagation of SPhP modes in the $SiO_2$ NRs and the reservoir. As a polar material, $SiO_2$ exhibits a positive or negative dielectric constant at different wavelengths due to the presence of optical phonons, which can show either dielectric or metallic behavior. Previous studies have shown that $SiO_2$ waveguides can support SPhP modes in both metallic and dielectric regions even beyond the optical phonon frequency range[9,18,19]. The modes in the dielectric region are also called Zenneck modes and optical guided modes. Here, we performed mode analysis on representative $SiO_2$ NRs with width of 10 μm and thicknesses of 50, 500, 1000 nm to obtain the propagation property of SPhP modes by using a finite element method-based

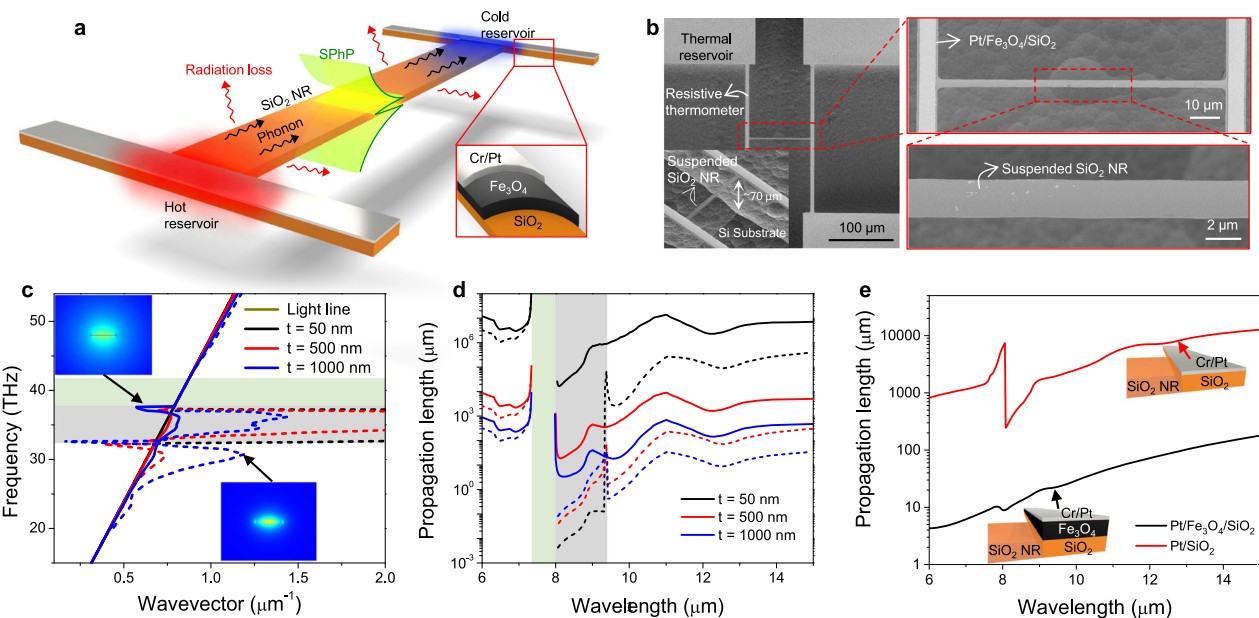

**Fig. 1 | SiO2 NR waveguide device and surface phonon polaritons (SPhP) properties. a** Schematic of heat conduction measurement of a $SiO_2$ NR waveguide, showing the heat conduction by SPhP and phonon in the waveguide between the hot and cold reservoirs, radiation loss from the waveguide, and direct background radiation between the two reservoirs. Each of these heat transfer pathways was measured and analyzed in this work to extract heat conduction mediated by SPhP in the waveguide. **b** SEM images of a representative $SiO_2$ NR device. Images of the SiO2 NR measured in this work are shown in Supplementary Fig.S2. **c** Dispersion relation and **d** Propagation length of SPhP modes supported on $SiO_2$ NRs. The $SiO_2$ NRs have rectangular cross-sections with width of 10 μm and thicknesses of 50, 500,

and 1000 nm. The solid and dash curves correspond to the quasi-TM and quasi-TE modes, respectively. The insets in (**c**) show the |E| field distributions of two representative modes. The gray area marks the metallic region, and the light green area marks the region with refractive index less than 1 where no valid propagation mode is supported. **e** Propagation length of SPhP along the thermal reservoir. The thermal reservoir supports two propagation modes (Supplementary Fig. S8) and here we focus on the mode which is primarily attributed to the conversion between SPhP and heat. The thickness of $SiO_2$, Pt and $Fe_3O_4$ are set to 44 nm, 110 nm and 350 nm, as in the experiments. The refractive indices are taken from literature[28,30,31].

commercial software (COMSOL Multiphysics). The solved dispersion relation and propagation length are shown in Fig. 1c and Fig. 1d, from which we can see that two modes are supported on SiO$_2$ NRs. In general, rectangular waveguides support hybrid quasi-TM (solid curve) and quasi-TE (dash curve) modes with the dominant polarization components along the thickness or width direction, respectively. Both modes have dispersion close to the light line, except for the quasi-TE mode in the metallic range (see Supplementary Note 7). The propagation length increases drastically with reduced waveguide thickness, because a larger portion of the mode volume is in the vacuum with zero loss. In the case of a 50 nm thick NR, the propagation length is greater than 1 cm, significantly larger than the MFP of phonons. This long propagation length is the main contributor to the high and ballistic SPhP thermal conductance in prior studies[8,9].

However, as mentioned earlier, the longer propagation length in a thinner NR is accompanied by a bigger mode size, as a larger portion of the wave is in the vacuum (see Supplementary Note 9). For the NR with width of 10 μm and thickness of 50 nm, the mode size at the wavelength of 6 μm is ~60 μm for quasi-TM mode and ~30 μm for quasi-TE mode. The large mode size poses two challenges for the observation of SPhP heat conduction in a NR waveguide: (i) the SPhP can be leaked to the substrate if the NR is not sufficiently separated from it; and (ii) the absorption of the wave with a large mode size by the thin reservoirs is very low without a proper design. By etching the Si underneath the NRs, we suspended the NRs from the Si substrate by at least ~70 μm to minimize the SPhP leakage (Fig. 1b). We then enhanced the absorption of SPhP by embedding a lossy dielectric layer of black oxide (Fe$_3$O$_4$) on the thermal reservoirs (inset of Fig. 1a and Supplementary Note 10). The black oxide greatly reduces the propagation length of the wave by about two orders of magnitude due to its lossy nature (Fig. 1e), thus enhancing the absorption efficiency.

## SPhP properties

Prior theoretical study has shown that the thermal conductance contributed by SPhP can be described as:[9]

$$G_{SPhP} = \frac{1}{2\pi} \sum_{n=1}^{N} \int_{\omega_n^{\min}}^{\omega_n^{\max}} \hbar \omega \frac{\partial f_\omega(T)}{\partial T} \tau_n(\omega) \eta_n(\omega) d\omega \qquad (1)$$

where $N$ is the number of modes, $\omega_n^{\min}$ and $\omega_n^{\max}$ stand for the lowest and highest frequencies of each mode. $f_\omega(T)$ is the Bose-Einstein distribution function and $T$ is the environmental temperature, $\tau_n$ is the transmission probability, and $\eta_n$ is the absorption efficiency by the reservoir. Here in our case, we take $N = 2$ for the two modes (quasi-TM and quasi-TE), and $\tau_n = 1$ because the propagation length is much longer than the physical size (ballistic). However, the absorption efficiency $\eta_n$ highly depends on the specific geometry and material properties, and computational complexity makes it difficult to make a quantitative calculation, considering the nanometer scale NR features and over 100 μm calculation domain size. Nevertheless, we can still use some physical principles to obtain its qualitative dependence trend and thus demonstrate the contributions of SPhP modes. The absorption efficiency $\eta_n$ is proportional to the coupling efficiency of the propagation wave from the SiO$_2$ NR to the sensing beam, which depends on the mode profile matching between the two structures (see Supplementary Note 9). For a SiO$_2$ NR, when the physical size is much smaller than the wavelength, the mode size ($MS$) is inversely proportional to the cross-section area ($A$) of the NR, i.e., $MS \propto 1/A$[18]. By substituting this into the thermal conductance equation above, we can obtain that $G_{SPhP} \propto A$ until it approaches a maximum value due to the saturation of the absorption efficiency.

## Measured apparent thermal conductivity of SiO$_2$ NRs

Figure 2 shows the measured apparent thermal conductivity of NR samples with various thicknesses and widths and at two different lengths ($L = $~50 and ~100 μm) within the temperature range of 100–550 K. Figure 2a shows that for the samples without the black oxide absorber, the measured thermal conductivity closely follows the bulk phonon thermal conductivity of SiO$_2$ for the entire temperature range regardless of the NR geometry ($t = 13$ and 21 nm). This suggests that the heat carried by SPhP in the NRs cannot be absorbed efficiently in the thermal reservoirs to contribute to the measured thermal conductivity. On the other hand, the measured thermal conductivity of NRs with similar geometries but with black oxide absorbers show enhanced thermal conductivity over the phonon limit. This is consistent with the expectation for polaritonic heat conduction. Figure 2b, c shows similarly enhanced thermal conductivity of NR samples with black oxide absorbers, with $t$ ~ 20 nm & $W$ ~ 1–2 μm (Fig. 2b) and $t$ ~ 16-50 nm and $W$ ~ 1–10 μm (Fig. 2c). For all the NR samples with black absorbers, this increase in thermal conductivity is larger at higher temperatures, consistent with the trend expected for heat conduction mediated by SPhP. The highest value is 2.14 W m$^{-1}$K$^{-1}$ at 550 K (or 34% higher than bulk SiO$_2$ thermal conductivity) for the sample of $t = 22$ nm, $W = 0.93$ μm, and $L = 49$ μm. This work provides a clear experimental observation of enhanced thermal conductivity contributed by SPhP through rational design of absorbers.

## SPhP thermal conductance of the SiO$_2$ NRs

We then extracted thermal conductance mediated by SPhP ($G_{SPhP}$) by analyzing and subtracting other components contributing to the measured total apparent thermal conductivity: (i) Phonon thermal conductance ($G_{ph}$) through the NRs, (ii) Radiation heat loss from the NR to the ambient, (iii) Direct background thermal conductance ($G_{bg}$) from the hot to cold thermal reservoirs via thermal radiation. The phonon contribution is calculated using the well-known thermal conductivity of amorphous SiO$_2$[20] and the NR geometry. The phonon thermal conductivity of amorphous SiO$_2$ slowly increases with temperature above 300 K and is independent of nanostructure size even down to 7.7 nm thickness as studied by us previously (SI in ref. 21), due to the short effective MFP of non-propagating diffusion (~1 nm) in amorphous SiO$_2$[22]. The radiation heat loss from the NR is analyzed along with the phonon heat conduction in the ribbon, manifested as progressively reduced apparent thermal conductivity on the ribbons with the same thickness but larger length (Fig. 2d). This "thermal fin" effect is due to the relatively larger radiation heat loss compared to heat conduction in a longer ribbon and has also been observed by us on SiO$_2$ NRs with larger thickness (~100 nm) without the black absorber design[23]. Following the same procedure in our prior work[23], we extract the emissivity ($\varepsilon$) of the SiO$_2$ NR by comparing the measured apparent thermal conductivity to the bulk value in the 400-μm-long NR. The extracted $\varepsilon$ value at different temperatures for the 400 μm long sample is shown in Supplementary Fig. S15: $\varepsilon$ ranges from 0.012 to 0.005 when temperature increases from 300 to 500 K. The colored area in Fig. 2d corresponds to the range of the extracted $\varepsilon$ value. This $\varepsilon$ is considerably lower than the bulk value of SiO$_2$ (~0.9) and that of thicker NRs studied earlier (~0.1 for $t = 100$ and $W = 11.5$ μm[23]) due to the much smaller thickness. To obtain $\varepsilon$ of NRs with different geometries (e.g., $t = 22$ nm and 44 nm in Fig. 2b, c), we also modeled $\varepsilon$ using full-wave simulations with COMSOL Multiphysics (see Supplementary Note 11). The modeled $\varepsilon$ for the 30-nm-thick NR agrees well with the measured value shown in Supplementary Fig. S15. We then used the model $\varepsilon$ to estimate the radiation heat loss from the NRs (Supplementary Note 12).

The third component, $G_{bg}$ is determined by measuring a blank device without a NR between the heating and sensing beams, as shown in Supplementary Fig. S16. The background conductance originates from radiation heat exchange between the two beams, as manifested by its rapid increase with temperature and exponential decrease with larger distance between the beams. The measured $G_{bg}$ at 550 K is significantly greater than the blackbody limit with the consideration of

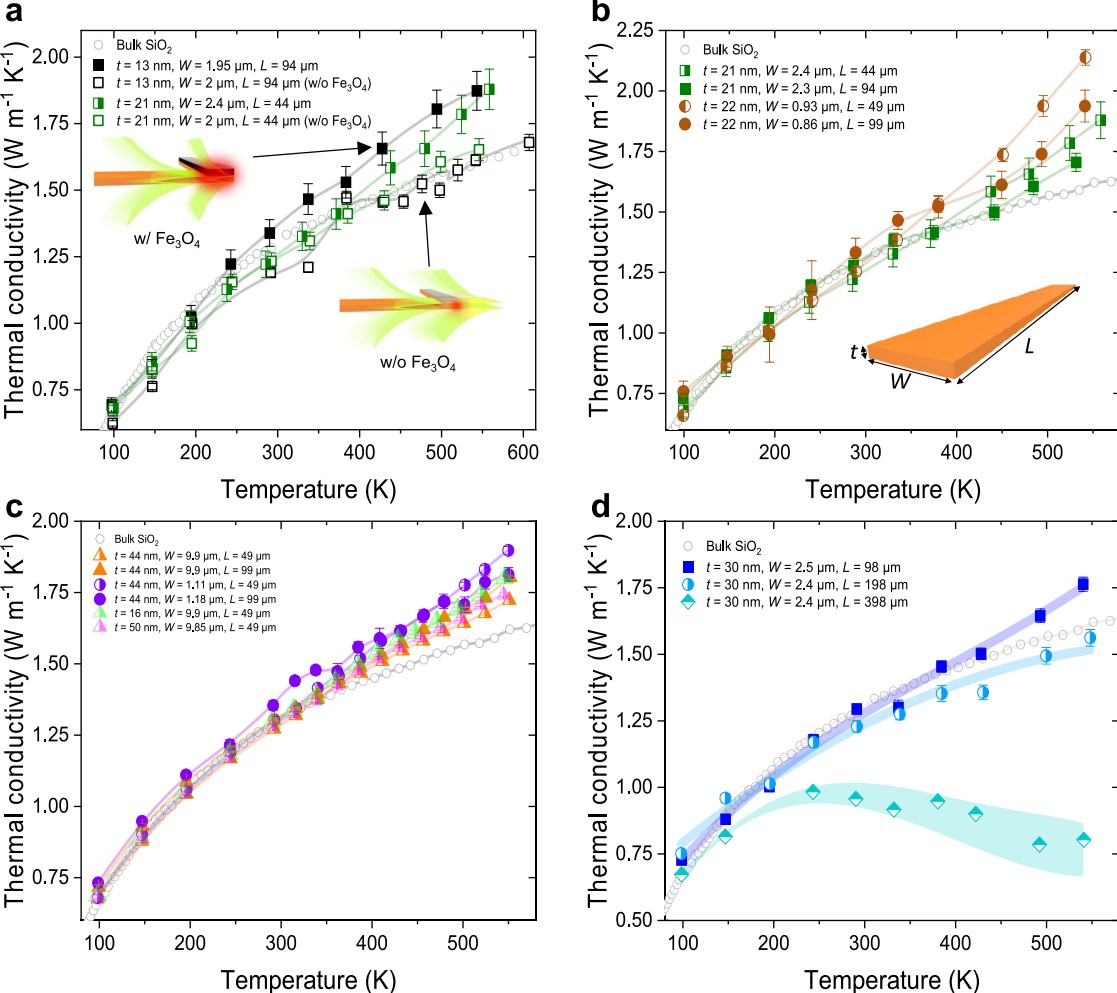

**Fig. 2 | Measured apparent thermal conductivity of SiO2 NRs with various geometries and with and without absorber designs. a** Comparison of thermal conductivity of samples with and without the $Fe_3O_4$ absorber. **b** the measured apparent thermal conductivity of NR samples with $t = 21–22$ nm. **c** The measured apparent thermal conductivity of NR samples with $W$ around 1 and 10 μm. **d** Apparent thermal conductivity of NR samples with similar width and thickness and different lengths, showing the radiation heat loss on long samples. The empty circles in all the figures are the bulk thermal conductivity value of amorphous $SiO_2$ taken from ref. 20. Error bars in all figures are determined by the standard deviations from the conductance measurements and NR's dimension determination, following previously published protocols[23,32]. We Conductance measurement error originates from the linear regression fitting applied to the temperature rise versus power plots (Supplementary Fig. S6). NR dimension determination error arises due to non-uniform width of NRs.

the view factor between the beams (see Supplementary Note 13). This is caused by the directional thermal radiation mediated by SPhP in the $SiO_2$ beams, as observed by ref. 11.

The extracted SPhP conductance (see Supplementary Note 14 for the extraction process) is shown in Fig. 3. Figure 3a shows $G_{SPhP}$ of each sample at ~550 K as a function of the sample length. We find that the sample with the larger thickness and width ($W = 9.9$ μm and $t = 44$ nm), $G_{SPhP}$ remains nearly constant over a wide length range of 49 μm to 99 μm, demonstrating the ballistic heat conduction behavior by SPhP. The ballistic transport regime over 50 μm at high temperature is expected for SPhP due to its larger wavelength and long propagation length (Fig. 1d) and is remarkable when compared to traditional heat carriers such as phonons that typically have MFP <~1 μm at room temperature[24–26]. On the other hand, the samples with smaller width and thickness show a reduction in $G_{SPhP}$ in longer NRs (Fig. 3a inset). This reduction is not due to diffusive thermal transport. Rather, the larger mode size of the SPhP waves in thinner and narrower NRs results in stronger leakage of the wave to the substrate and, consequently less efficient coupling to the thermal reservoirs. This can be clearly seen from the SPhP waveguiding mode profile along the thickness direction of two NR samples with different thickness and width, as depicted in

Fig. 3b. As mentioned earlier, the majority of the mode volume of the SPhP wave is in vacuum. The mode size of SPhP increases rapidly with decreasing ribbon width and thickness (Supplementary Note 9). For the NR with ~1 μm width and 21 nm thickness, the mode size is around 400 μm, which is much larger than the gap between the $SiO_2$ NR and the Si substrate (~70 μm), resulting in significant energy leak to the substrate before reaching the sensing reservoir. As the thickness and width of the $SiO_2$ NR increase, the SPhP mode is increasingly confined to the vicinity of the NR (mode size ~40 μm when $W = 9.9$ μm and $t = 44$ nm), resulting in negligible leakage to the substrate during the propagation along the NR. It can be seen from Fig. 3c that the SPhP contributed thermal conductance of 16 nm thick sample (~0.5 nW/K at 550 K) is lower than 50 nm one (~0.8 nW/K) due to the large loss. The two samples have similar width and length, but the 16 nm one has much smaller thickness, its thermal conductivity is higher (Fig. 2c) due to the normalization of their cross sections: $k = \frac{G \cdot L}{W \cdot t}$.

## Discussion

The large wavelength of SPhP makes it possible to observe the low-dimensional heat conduction behavior with SPhP that is otherwise only accessible in exotic phononic or electronic heat conduction systems,

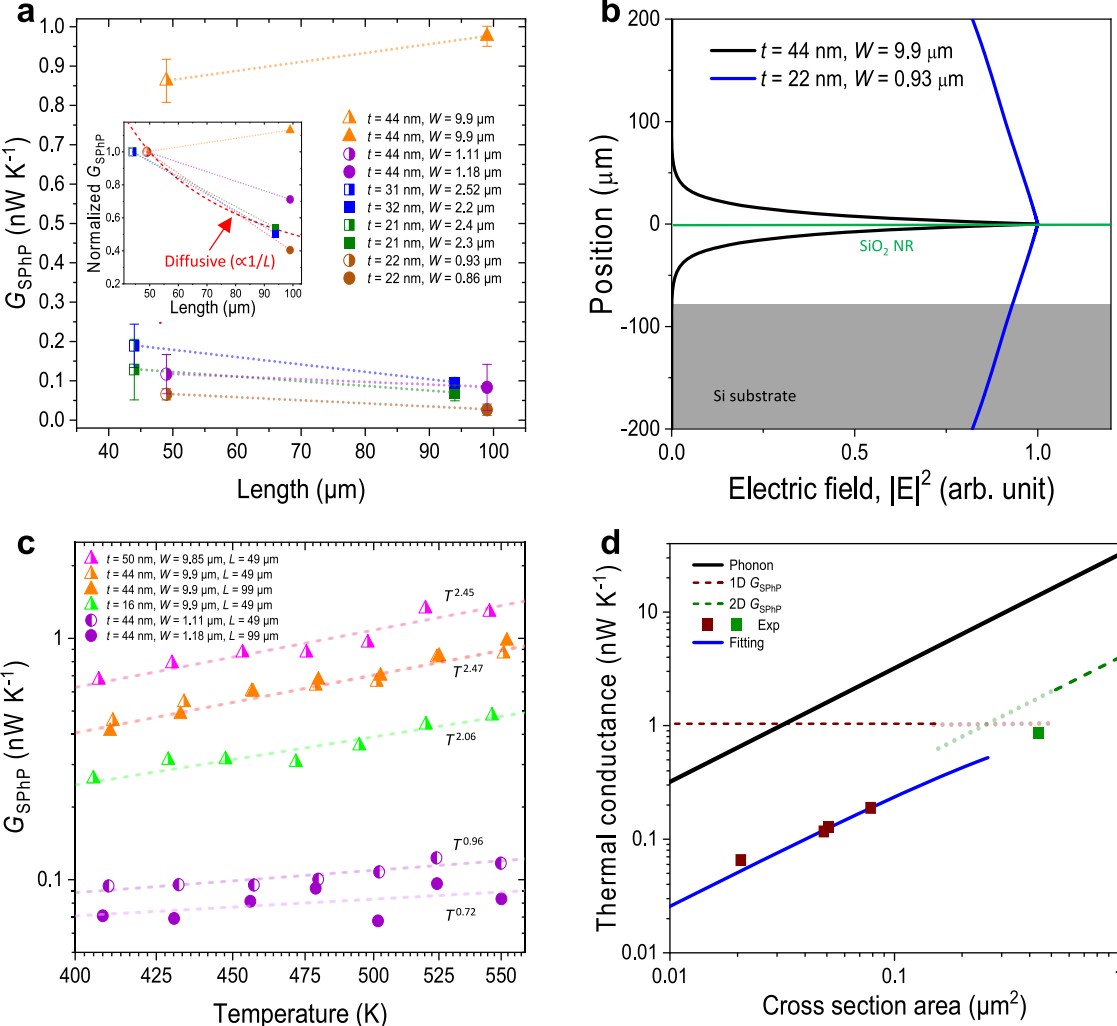

**Fig. 3 | Ballistic and low-dimensional SPhP thermal conductance. a** Length-dependence of SPhP thermal conductance ($G_{SPhP}$) of the SiO$_2$ NRs at 550 K. Inset: normalized $G_{SPhP}$ with the value of ~50 μm long samples. **b** Simulated mode profile of the SiO$_2$ NR waveguide quasi-TM mode at the wavelength of 10 μm in the thickness direction for two samples: $t = 22$ nm and $W = 0.93$ μm (blue) and $t = 44$ nm and $W = 9.9$ μm (black). The absolute value of electric field intensity for blue line is much smaller than black one. The intensity is normalized with their maximum value in the figure for better comparison. The thinner and narrower the NR, the larger the mode size, leading to SPhP energy leakage to the substrate and difficult to coupling to reservoir. **c** Temperature-dependent $G_{SPhP}$ of the SiO$_2$ NRs

with different thickness and different widths of ~1 and ~10 μm, showing close to $T$ and $T^2$ dependence respectively. The apparent thermal conductivity of these samples is shown in Fig. 2c. **d** Dependence of $G_{SPhP}$ on the NR cross-section area compared to the calculated phonon thermal conductance of 50-μm-long NRs (solid black line) and the 1D and 2D quantum limits of $G_{SPhP}$ (red and green dashed lines) in the small and large cross-section area regimes respectively. The experiment data corresponding to the ~50-μm-long NRs in (**a**). Red dots for samples with small cross section (1D), green dot for sample with larger width (2D). The blue solid line is the fitting of the experimental data (red dots) to the 1D theoretical $G_{SPhP}$ using Eq. (1). All the values shown in (**d**) are for 550 K.

such as atomic and molecular structures and sub-Kevin phononic waveguide. For NRs with thickness and width smaller than the wavelength (~5 μm at 550 K), the transport should be one dimensional. Similarly, for NRs with thickness smaller than the wavelength but width comparable to the wavelength, the transport is close to two-dimensional. We fit the measured $G_{SPhP}$ as a power law function of temperature ($G_{SPhP} \sim T^\alpha$), and find that $\alpha$ is close to 1 and 2 for the ~1 μm and ~10 μm wide samples, respectively within the temperature range of 400–550 K. (Fig. 3c). This is consistent with the theoretical prediction of 1D and 2D $G_{SPhP}$ in narrow and wide SPhP waveguide, respectively[9,27]. However, the exponents $\alpha$ deviate from 1 or 2. This is due to the fact that the observed $G_{SPhP}$ is still only a fraction of the respective theoretical 1D and 2D limits. For example, $G_{SPhP}$ of the ~1 μm wide samples is about 10% of the theoretical 1D limit. The deviations from the theoretical limit are caused by the losses of SPhP waves, either leaked into the substrate during propagating or scattered into free space when coupling into the cold reservoir. Both loss

mechanisms depend on the mode sizes of the waves: the larger the mode size, the bigger the losses. The mode size mainly depends on the cross section of the sample and it is larger for samples with smaller cross sections. That is the reason for the lower $G_{SPhP}$ in the ~1 μm wide samples. For a specific sample, the mode size could also change with the temperature. This is mainly because the mode size depends on the wavelength, and the peak thermal wavelength of the emission spectrum depends on the temperature. However, for a narrow temperature range such as from 400 to 550 K, the peak thermal wavelength does not change dramatically. Furthermore, the spectral dependence of the mode size is also quite weak for the case considered here. For SiO$_2$, the material property varies dramatically within the Reststrahlen band but changes slowly with wavelength outside the Reststrahlen band[28], so does the mode size as shown in Supplementary Note 9. The modes on the SiO$_2$ NRs are supported with frequency range much broader than the Reststrahlen band, so the average mode size does not change a lot with temperature especially at relatively high temperatures. That

means the percentage of measured $G_{SPhP}$ relative to the theoretical value is almost the same, so the measured $G_{SPhP}$ also show similar $T$ dependence as the theoretical prediction within the narrow temperature range studied.

In an SPhP waveguide, when the wavelength of SPhP is larger than the characteristic sizes of samples along two or one directions and with perfect transmission and absorption coefficient (i.e., $\tau_n = 1$ and $\eta_n = 1$ in Eq. 1), 1D and 2D quantum limits of thermal conductance can be achieved[9,27]. The 1D quantum limit of $G_{SPhP}$ is proportional to $T$ and is only dependent on physical constants[9]. The 2D limit is proportional to $T^2$ and the ratio $G_{SPhP}/W$ only depends on physical constants[22]. In Fig. 3d, we plot $G_{SPhP}$ at 550 K as a function of the cross-section area of the NRs ($A = tW$) and compare it to the 1D and 2D quantum thermal conductance limit (the thickness and width dependence is shown in Supplementary Note 15). We observed that $G_{SPhP}$ is still lower than the quantum limits due to the non-ideal absorption to the reservoirs. We also found that $G_{SPhP}$ increases with a larger cross-section area due to the smaller mode size, less energy leakage, and better absorption. We fit the experimental data using a simplified relation to account for the non-ideal absorption coefficient, $\eta_n \approx \tanh(A/C_{eff})$, shown as the blue curve in Fig. 3d (more details in Supplementary Note 9). A single fitting parameter $C_{eff} = 0.45 \mu m^2$ can fit all the experimental data, showing the validity of our physical picture. This result suggests that one could realize the quantum limit of thermal conductance at room and high temperatures if the absorption coefficient can be enhanced to near unity, for instance, by using more optimized absorber materials and geometry.

The observed $G_{SPhP}$ is still lower than the phonon limit (see Fig. 3d). However, as the phonon conductance follows the Fourier's law, namely, proportional to the cross-section area and inversely proportional to the length, one can potentially realize a regime where $G_{SPhP}$ is much larger than the phonon limit if the absorption coefficient can be optimized to near the unity. For the SiO$_2$ NRs with the 50–100 μm length studied here, this regime can be achieved when the cross-section area is less than 0.03 μm$^2$ (i.e., the regime above the black curve and below the red curve in Fig. 3d). For samples with larger cross-section area (such as the wide ribbons), one could also maintain high $G_{SPhP}$ but lower phonon conductance in NRs with nanoscale porosity in it, or with very long NRs due to the ballistic nature of SPhP, provided that the mode size is substantially smaller than the gap between the NR and the substrate. Finally, hyperbolic metamaterial with high-density metal-dielectric interfaces[29] is also a promising system to observe high $G_{SPhP}$ as each interface can support SPhP modes.

In summary, we have experimentally realized enhanced thermal conductivity mediated by SPhP in a polar dielectric (SiO$_2$) nanoribbon waveguide, which represents an entirely new heat conduction regime that has been pursued for nearly two decades. The new regime shows combined characteristics of text-book heat conduction (high energy flux) and thermal radiation (ballistic and large wavelength). We identified the key factors that are crucial for the measurement of heat flux carried by SPhP but have been overlooked in prior theoretical predictions and experimental attempts: optimal mode size surrounding the waveguide and efficient absorption of the wave by the thermal reservoirs. We then carefully designed our NR waveguide geometry and the absorbers in the reservoirs to realize an overall thermal conductivity greater than the well-known phonon limit in amorphous SiO$_2$. Due to the large wavelength of SPhP, the SPhP thermal conductance shows 1D and 2D temperature dependence in narrow and wide NRs, respectively even at room and high temperatures. For the sample that has little leakage of SPhP to the substrate, we showed length-independent, ballistic thermal conductance. This work paves the way for observing low-dimensional heat conduction phenomena mediated by SPhP waves and could lead to the realization of quantum thermal conductance at high temperature as well as high SPhP thermal conductance over the phonon limit, and thus could have broad implications in fields such as microelectronics and nanophononics.

## Methods

### Device fabrication

SiO$_2$ NRs were fabricated from a 100-nm thick thermally grown SiO$_2$ layer on a Si wafer. The Fe$_3$O$_4$/Pt beams were deposited and patterned by sputtering and liftoff. The SiO$_2$ NRs and the SiO$_2$ underneath the beams were defined by photolithography and reactive ion etching (RIE). The Si substrate outside the NR and beam regions was etched using deep RIE. The thickness of the SiO$_2$ NRs was then controllably reduced by etching in a buffered oxide etchant. Finally, the NRs and the beams were suspended from the substrate using dry XeF$_2$ etching. The fabrication process is shown in detail in Supplementary Note 1. There is a small but non-negligible etching of SiO$_2$ by XeF$_2$ (Supplementary Note 3). The thickness of the obtained SiO$_2$ NR was determined from both AFM imaging and frequency-dependent thermal transport measurements, which yielded the heat capacity and, hence the thickness of the NR sample[16,17] (Supplementary Note 6).

### Thermal conductivity measurement

We employed high-resolution heat conduction measurement (~10 pW K$^{-1}$) using a modulated heating technique and a noise-canceling bridge scheme, as demonstrated in our earlier work[15]. The Pt layers on the two SiO$_2$/Fe$_3$O$_4$/Pt beams were used for joule heating on the heating beam and for thermometry on both beams. An AC voltage ($V_{1\omega}$) at a modulated frequency $\omega$ was applied to the heating beam. $3\omega$ voltage of the heating beam was measured to determine the temperature rise and the thermal conductance of the beam. The sensing beam forms a Wheatstone bridge with a pairing beam of the similar resistance inside the same cryogenic vacuum chamber, along with two external resistors. A small DC current was applied to the bridge and $2\omega$ voltage was measured from the bridge, which was used to determine the temperature rise on the sensing beam and ultimately the thermal conductance of the SiO$_2$ NR. A low frequency (e.g., ~0.5 Hz) was chosen to ensure that the thermal penetration length was longer than the NRs and the beams. The modulated heating and the Wheatstone bridge were used to greatly suppress the effects of the $1/f$ noise and environmental temperature fluctuation. The detailed description of the measurement setup and principle can be found in Supplementary Notes 4 and 5.

### Electromagnetic modeling

The characteristics of the SPhP modes in the SiO$_2$ NR, including the dispersion relation, propagation length and the mode profile were obtained from numerical simulations using a mode analysis module in a finite element software (COMSOL Multiphysics). The SiO$_2$ waveguide was modeled as a rectangular shape with width of $W$, thickness $t$ with the surrounding medium as vacuum. The refractive indices of SiO$_2$, Fe$_3$O$_4$, Pt were obtained from literature[28,30,31]. Both the quasi-TM and quasi-TE modes were obtained from the simulations. The theoretical limit of the thermal conductance due to SPhP was calculated using the Landauer formula (Eq. 1) by assuming 100% transmission. Details of the modeling can be found in Supplementary Notes 7–9. The surface emissivity of the SiO$_2$ NR was obtained by performing full-wave simulations in COMSOL Multiphysics. The SiO$_2$ NR was set to be infinitely long using periodic boundary conditions at two ends and surrounded by vacuum with a perfect matching layer as the boundary condition to ensure accuracy. The incident light was modeled as a plane wave with a certain angle of incidence ($\alpha,\theta$) and the absorption cross-section was then calculated. The emissivity was obtained using the Kirchhoff's law (Supplementary Note 11).

## Data availability

The data that support the findings of this study are available within the paper and the Supplementary Information. Other relevant data are available from the corresponding authors on reasonable request.

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

## Acknowledgements

This work is supported by the Condensed Matter Physics Program of the Division of Materials Research of the U.S. National Science Foundation (Grant No. DMR-2005181, to R.C. and Z.L.). This work was performed in part at the San Diego Nanotechnology Infrastructure (SDNI) of UCSD, a member of the National Nanotechnology Coordinated Infrastructure, which is supported by the National Science Foundation (Grant no. ECCS-2025752).

## Author contributions

R.C. and Z.L. conceived the idea and supervised the project. Y.P. fabricated the devices. L.C. performed the EM simulations. Y.P. and W.J. did the thermal transport measurements and analyzed the data. All the authors contributed to the writing and editing of the manuscript.

## Competing interests

The authors declare no competing interests.
