## [Peer Review File · Nature Communications]

Low-dimensional Heat Conduction in Surface Phonon Polariton WaveguideREVIEWER COMMENTS

Reviewer #1 (Remarks to the Author):

The paper by Pei et al. experimentally presents an enhanced thermal conductivity mediated by SPhP (surface phonon polaritons) in SiO₂ nanoribbon waveguides. SPhP-mediated heat transfer has been predicted theoretically two decades ago, but the direct observation of the increased thermal conductivity by SPhP has not reported yet due to the difficulty in fabricating large-scale SiO₂ membrane, which is crucial to make SPhP-mediated thermal conductivity be comparable to that by phonon. In this manuscript, authors carefully design SiO₂ nanoribbon waveguides to make phononic and SPhP contributions to heat conductance be comparable. As a result, they could observe 34% increased thermal conductivity in nanoribbons of SiO₂. As authors addressed, this is the first work which shows increased heat transfer mediated by SPhP, although recent paper reported an enhanced heat transfer by SPP (surface plasmon polaritons), which was not mentioned in this manuscript.

Unlike prior studies, which tried to measure increased heat dissipation by SPhPs and SPPs, the authors measured temperature rises at two points. To ensure that heat flux by SPhP actually affects temperature difference between two points, absorption of the heat carried by SPhP at the cold reservoir is crucial and authors introduced Fe₃O₄ to achieve an efficient coupling between SPhP mode and cold reservoirs. It is obvious that the authors observed increased thermal conductivity by SPhP, because they demonstrated clear deviation between thermal conductivities with and without Fe₃O₄. From this point of view, I think this work is quite interesting and original. However, I have a few technical concerns which should be addressed before publication.

1. In Eq. (1), what is the exact value of integration range? In other words, please specify lowest and highest frequencies for each mode. It seems that the author takes lowest frequency as 0 and highest frequency as infinity as written in Supplementary Note 7. However, according to L. Tranchant et al (Nano Lett. 2019, 19, 6924-6930), they set highest frequency as 258 Trad/s (~ 7.3 μ m) for silica because all of the TM-guided modes with frequencies higher than 258 Trad/s cannot be thermally excited. It can be optically excited, but these modes are not related to a phonon-photon coupling. In page 13 of manuscript, wavelength of 5 μ m (higher frequency than 258 Trad/s) has been discussed. The authors should clarify the exact range of integration range and provide the basis.
2. When deriving absorption efficiency by the reservoir, the authors assume that all the energy coupled to the sensing beam is eventually absorbed. To back up the assumption, they show greatly reduced propagation length when Fe₃O₄ is introduced. However, in Fig. S9, there are two modes which can be supported in Pt/Fe₃O₄/SiO₂ structure. And one of the modes has long propagation length, meaning that it cannot be absorbed well like the case of Pt/SiO₂ reservoir. It is quite confusing because there are also two modes for SiO₂ nanoribbons. The authors should clarify how two modes in SiO₂ nanoribbons can be coupled to two modes in reservoirs.
3. In Supplementary Note 12, when calculating radiation heat loss, it seems that nanoribbons are assumed to be solely surrounded by enclosure (refer to Eq. S22). However, there exists large Si substrate below nanoribbons and Si substrate is at an elevated temperature. Is the effect of Si substrate negligible?
4. When deriving the value of emissivity, the author assumes infinitely long nanoribbons. Is

there any dependency of emissivity on the length of nanoribbons? What do you mean by 'long propagating modes' in page 20 of Supplementary Note.

5. When calculating Bose-Einstein distribution function, which value did you use as reference temperature? Is it mean temperature between hot and cold reservoirs?

6. When plotting Fig. 3b, what is the reference wavelength?

7. For Fig. 3c, the authors argue that 1- μm -wide sample follows the T dependence. However, because it has very small cross section, the mode size should be very large, leading to significant leakage to substrate. Is it still reasonable that it is 1D heat conduction, which follows the T dependence, although there should be a large deviation between theoretical prediction and experimental result?

8. In this manuscript, there are many factors which can affect both theoretical predictions and experimentally derived value of SPhP-mediated thermal conductivity. For example, when obtaining absorption efficiency, C is related to frequency and material property. The experimental data in Fig. 3d are obtained with nanoribbons with different cross section. Their mode size should be a function of wavelength and those nanoribbons should have different cut-off wavelength (i.e., wavelength wo leakage) because the gap between nanoribbons and Si substrate is 70 μm . However, the experimental data with different cross-sectional area are all fitted to single fitting parameter $C = 0.45 \mu\text{m}^2$. How is it possible and what is the physical meaning of this constant fitting parameter? If you could provide more detailed explanation, it would help general readers' understanding.

Reviewer #2 (Remarks to the Author):

The authors investigated the enhanced thermal conductivity due to surface phonon polaritons (SPhPs) in SiO₂ nanoribbon waveguides. They measured thermal conductivities of different samples with different structural parameters and with/without Fe₃O₄ absorber. The comparison clearly shows the contribution of SPhPs to the thermal conductivity. The authors carefully design the structure and examine the data to demonstrate SPhP thermal conduction. The level of sample fabrication and measurement is very high. The manuscript is well written and contains useful information for the heat transfer community. However, some of the authors' claims need careful consideration and there are several data deficiencies. This manuscript may be reviewed and considered for publication after the revision.

*In p3, "Our results are the first time to demonstrate an enhanced effective thermal conductivity due to SPhP..."

Ref. 13 reported an enhanced thermal conductivity due to SPhP. Therefore, this claim is incorrect. The authors need to limit the novelty to the correct range and describe it accurately without overextending it. Also this claim should be updated throughout the manuscript.

* In the abstract, the authors wrote "direct observation". What does direct mean? The authors compare the thermal conductivity with and without an absorber, which can be called indirect. Ref. 13 compares the thermal conductivities at different temperatures to prove that SPhPs are heat carriers. I think both are at similar levels. Also, "direct" appears on the

bottom line of p. 8.

* Why are there no error bars in Figure 2a, although there are in other figures? Please also clarify how the error bars are determined in the manuscript.

*The authors schematically draw the mode profile of the waveguide modes in membranes in Fig. 3b. The authors should calculate and show the real mode profile to claim that the significant energy leakage to the substrate in a thinner membrane.

* If there is more leakage to the substrate in a thinner membrane, why is the thermal conductivity for $t = 16$ nm (green) higher than that for $t = 50$ nm (orange) in Fig. 2c?

* In Fig. 3b the authors draw lines of T^1 and T^2 for the small W and large W structures respectively. It misleads the analysis and is not as convincing to conclude that these are 1D and 2D Gsphp. The authors should fit the experimental data and obtain the value of α (T^α) and discuss whether these are close to 1 or 2. Then please conclude that these are 1D or 2D Gsphp.

Responses to Reviewers' Comments

Reviewer #1 (Remarks to the Author):

The paper by Pei et al. experimentally presents an enhanced thermal conductivity mediated by SPhP (surface phonon polaritons) in SiO₂ nanoribbon waveguides. SPhP-mediated heat transfer has been predicted theoretically two decades ago, but the direct observation of the increased thermal conductivity by SPhP has not reported yet due to the difficulty in fabricating large-scale SiO₂ membrane, which is crucial to make SPhP-mediated thermal conductivity be comparable to that by phonon. In this manuscript, authors carefully design SiO₂ nanoribbon waveguides to make phononic and SPhP contributions to heat conductance be comparable. As a result, they could observe 34% increased thermal conductivity in nanoribbons of SiO₂. As authors addressed, this is the first work which shows increased heat transfer mediated by SPhP, although recent paper reported an enhanced heat transfer by SPP (surface plasmon polaritons), which was not mentioned in this manuscript.

Unlike prior studies, which tried to measure increased heat dissipation by SPhPs and SPPs, the authors measured temperature rises at two points. To ensure that heat flux by SPhP actually affects temperature difference between two points, absorption of the heat carried by SPhP at the cold reservoir is crucial and authors introduced Fe₃O₄ to achieve an efficient coupling between SPhP mode and cold reservoirs. It is obvious that the authors observed increased thermal conductivity by SPhP, because they demonstrated clear deviation between thermal conductivities with and without Fe₃O₄. From this point of view, I think this work is quite interesting and original. However, I have a few technical concerns which should be addressed before publication.

Response:

We thank this reviewer for the strong recommendation and for recognizing our work as “interesting and original”. We also thank the reviewer for many helpful and constructive comments, which we have carefully addressed in details, as shown below.

1. In Eq. (1), what is the exact value of integration range? In other words, please specify lowest and highest frequencies for each mode. It seems that the author takes lowest frequency as 0 and highest frequency as infinity as written in Supplementary Note 7. However, according to L. Tranchant et al (Nano Lett. 2019, 19, 6924-6930), they set highest frequency as 258 Trad/s ($\sim 7.3 \mu\text{m}$) for silica because all of the TM-guided modes with frequencies higher than 258 Trad/s cannot be thermally excited. It can be optically excited, but these modes are not related to a phonon-photon coupling. In page 13 of manuscript, wavelength of 5 μm (higher frequency than 258 Trad/s) has been discussed. The authors should clarify the exact range of integration range and provide the basis.

Response:

The integration range over frequency from 0 to infinity is considered in this work as the ideal case for the theoretical maximum limit.

We agree with the reviewer that the guided mode above the optical phonon cutoff frequency cannot be thermally excited in systems containing only SiO₂, which is the case in L. Tranchant et al (Nano Lett. 2019, 19, 6924-6930). However, in our experimental system, we have the heating and sensing beam with Pt and Fe₃O₄ included, which will contribute to the mode excitations and absorption beyond this frequency and thus contribute to thermal conductance. Considering that both Pt and Fe₃O₄ are lossy at a broadband frequency covering the whole range of Bose-Einstein distribution at our interested temperature, we use the integration range from 0 to infinity. Similar thermal excitation beyond the optical phonon range was already reported [Ref: Gluchko, S., Palpant, B., Volz, S., Braive, R., & Antoni, T. . *Applied Physics Letters*, 110, 263108 (2017)]. In that paper, a broadband thermal excitation up to 3725 cm⁻¹ (wavelength of 2.7 μm) was experimentally demonstrated on SiO₂ submicron film.

We have made the following modifications:

1. the main text on page 6 as follow:

“Previous studies have shown that SiO₂ waveguide can support SPhP modes in both metallic and dielectric regions even beyond the optical phonon frequency range^{9,21,22}. The modes in the dielectric region are also called Zenneck modes and optical guided modes.”

2. add the citation “Appl. Phys. Lett. 110, 263108 (2017)” as the experimental demonstration of thermal excitation of SPhP beyond the optical phonon range .
3. Supplementary Note 7 on Page 12 as follows.

“The value of ω_n^{min} and ω_n^{max} are considered as the lower and upper frequency limit of optical phonon of the materials. For example, 7.6 Trad/s and 258 Trad/s are used for SiO₂ in literatures⁴. In experiments like our system, the heating and sensing beam with Pt and Fe₃O₄ are included, which will contribute to the broadband mode excitations and absorption and thus contribute to thermal conductance. Considering that both Pt and Fe₃O₄ are lossy at a broadband frequency covering the whole range of Bose-Einstein distribution at our interested temperature, we assume $\omega_n^{min} = 0$, $\omega_n^{max} = \infty$. Similar thermal excitation beyond the optical phonon range was already reported.⁵”

2. When deriving absorption efficiency by the reservoir, the authors assume that all the energy coupled to the sensing beam is eventually absorbed. To back up the assumption, they show greatly reduced propagation length when Fe₃O₄ is introduced. However, in Fig. S9, there are two modes which can be supported in Pt/Fe₃O₄/SiO₂ structure. And one of the modes has long propagation length, meaning that it cannot be absorbed well like the case of Pt/SiO₂ reservoir. It is quite confusing because there are also two modes for SiO₂ nanoribbons. The authors should clarify how two modes in SiO₂ nanoribbons can be coupled to two modes in reservoirs.

Response:

Thanks for the question. Each of the two modes on the SiO₂ nanoribbons can be coupled to either one of the two modes on the reservoirs, i.e., four possible coupling pathways, which are marked as A, B, C and D as shown in the schematic figure below. The coupling efficiency, in principle, can be estimated by using mode coupling theory as we include as the equation S14 in Supplementary Note 9.

$$\eta = \frac{\omega \epsilon_0 \int_{-\infty}^{\infty} \int_{-\infty}^{\infty} (n_1^2 - n_2^2) E_1^* \cdot E_2 dx dy}{\int_{-\infty}^{\infty} \int_{-\infty}^{\infty} n_z \cdot (E_1^* \times H_1 + E_1 \times H_1^*) dx dy}$$

Where n_1 , E_1 and H_1 correspond to the mode profile of either mode on SiO₂ nanoribbons, n_2 and E_2 correspond to the mode profile of either mode on the reservoirs. The total coupling efficiency from quasi-TM mode and quasi-TE mode are $\eta_{quasi-TM} = \eta_A + \eta_B$ and $\eta_{quasi-TE} = \eta_C + \eta_D$. The terms η_A and η_C correspond to the couplings to the short propagation mode on reservoir and end up with high absorption and high coupling efficiency when the mode size on SiO₂ NR is small. The terms η_B and η_D correspond to the couplings to the long propagation mode on reservoir, and as the reviewer pointed out that they cannot be absorbed well. In our analysis, we neglect the contributions from η_B and η_D . We simply use η_A and η_C to represent the contributions from quasi-TM and quasi-TE modes to thermal conductions (marked as the red arrows in the figure). The actual efficiency is spatial, structural and wavelength dependent, which makes the numerical calculations very challenging. Here we mainly use theoretical relations as a guidance to improve the overall efficiency in our experimental design.

Figure R1 (Fig. S12 in the revised SI): Possible mode couplings from the SiO₂ NR to the sensing beam (reservoir). Each of the two modes on the SiO₂ nanoribbons can be coupled to either one of the two modes on the reservoirs, i.e., four possible coupling pathways which are marked as A, B, C and D, respectively. The red line arrows mark the two couplings that make the dominant contribution to the thermal conductance. The inset in the dash line box shows the cross section of our structure.

We have modified **Page 16 of Supplementary Note 7** as follows for clarifying.

“The final absorption efficiency depends on the leakage during propagation, coupling efficiency from the SiO₂ NR to the sensing beam, and how much of this energy is eventually absorbed as heat. The SiO₂ NRs support two propagation modes as quasi-TM mode and quasi-TE mode. The sensing beam also supports two modes, one with a long propagation length and a large mode size, the other with a short propagation length and a small mode size (see Fig. S9). Each of the two modes on the SiO₂ NR can be coupled to either one of the two modes on the sensing beam, i.e., four possible coupling pathways, as shown as A, B, C and D in Fig. S12. The total coupling efficiency from quasi-TM mode and quasi-TE mode are $\eta_{quasi-TM} = \eta_A + \eta_B$ and $\eta_{quasi-TE} = \eta_C + \eta_D$. The terms η_A and η_C correspond to the couplings to the short propagation mode on sensing beam and end up with high absorption and high coupling efficiency when the mode size on SiO₂ NR is small. The terms η_B and η_D correspond to the couplings to the long propagation mode on sensing beam and contribute less to the thermal conductance due to low absorption efficiency. We can neglect the contributions from η_B and η_D and simply use η_A and η_C to represent the contributions from quasi-TM and quasi-TE modes to the thermal conduction (marked as the red arrows in the Fig. S12), i.e., $\eta_{quasi-TM} \approx \eta_A$ and $\eta_{quasi-TE} \approx \eta_C$. Please note that, from the coupling efficiency point of view, these four possible couplings have different coupling efficiencies depending on the mode similarity coefficient. For example, the SiO₂ NR with a large cross section has a smaller mode size and is more likely to couple to the mode on sensing beam with a small mode size, i.e., the couplings corresponding to the red arrows in Fig. S12. Considering that after we added a lossy Fe₃O₄ layer on the heating and sensing beam to enhance the absorption efficiency, the propagation length on the sensing beam reduced to a few micrometers to tens of micrometers, we assume that all the energy coupled to the short propagation mode on sensing beam is eventually absorbed.”

3. In Supplementary Note 12, when calculating radiation heat loss, it seems that nanoribbons are assumed to be solely surrounded by enclosure (refer to Eq. S22). However, there exists large Si substrate below nanoribbons and Si substrate is at an elevated temperature. Is the effect of Si substrate negligible?

Response:

Yes, the Si substrate effect is negligible due to the following two reasons.

First, during the measurement, the whole chamber environment including the Si substrate is carefully controlled at the same environmental temperature T , which varied from 300 K to about 550 K in this study. Only the hot reservoir of the device is heated up by Joule heating for thermal conduction measurement. In experiments, the temperature of the hot reservoir is $T_h = T + \Delta T_h$ and that of the cold reservoir is $T_s = T + \Delta T_s$, as shown in Supplementary Note 5. Therefore, we can consider the ribbon is being surrounded by an enclosure at the same temperature (T).

Second, since the ribbon is much smaller than the surrounding objects, include the vacuum chamber wall and the Si substrate, we can assume the surrounding enclosure as a blackbody, regardless of the actual optical properties or emissivity of the surrounding materials. Radiative heat transfer flux between two gray surfaces is [Ref: Bergman, T. L., Lavine, A. S., Incropera, F. P., & DeWitt, D. P. (2011). *Introduction to heat transfer*. John Wiley & Sons.]:

$$\dot{Q}_{12} = \frac{\sigma A_1 (T_1^4 - T_2^4)}{\frac{1-\varepsilon_1}{\varepsilon_1} + \frac{1}{F_{12}} + \frac{A_1(1-\varepsilon_2)}{A_2 \varepsilon_2}} \quad (\text{Eq. R1})$$

where A and ε are the surface area and emissivity respectively, σ is the Stefan–Boltzmann constant. Subscripts 1 and 2 means the objects 1 and 2. In our case of a very small nanoribbon (object 1) surrounded by a much larger enclosure (object 2), $F_{12} = 1$, and $\frac{A_1}{A_2} \cong 0$. The emissivity of nanoribbon ε_1 is very small. Therefore, the above equation is reduced to:

$$\dot{Q}_{12} = \varepsilon_1 A_1 \sigma (T_1^4 - T_2^4) \quad (\text{Eq. R2})$$

We can see that the radiative flux only depends on the emissivity of object 1 (i.e., nanoribbon in our case) and is independent of the properties of the enclosure (either the vacuum chamber wall or the Si substrate).

As mentioned earlier, $T_2 = T$ (enclosure or environmental temperature), $T_1 = T + \Delta T$ where $\Delta T \ll T$ (Note: ΔT is between ΔT_h and ΔT_s), so the radiative heat transfer coefficient (h_r) can be written as:

$$h_r = \frac{\varepsilon_1 A_1 \sigma (T_1^4 - T_2^4)}{T_1 - T_2} \cong 4 \varepsilon_1 \sigma T^3 \quad (\text{Eq. R3})$$

This is Equation S22 for h_r in Supplementary Note 22, which shows that h_r is independent of the detailed properties of the surrounding.

We have modified **Page 23 of Supplementary Note 12** as follows for clarifying.

“During the measurement, the whole chamber environment including the Si substrate is carefully controlled at the same environmental temperature T , which varied from 300 K to about 550 K in this study. Therefore, we can consider the ribbon is being surrounded by an enclosure at the same temperature (T). Since the ribbon is much smaller than the surrounding objects, include the vacuum chamber wall and the Si substrate, we can assume the surrounding enclosure as a blackbody, regardless of the actual optical properties or emissivity of the surrounding materials. Radiative heat transfer flux between two gray surfaces is¹¹:

$$\dot{Q}_{12} = \frac{\sigma A_1 (T_1^4 - T_2^4)}{\frac{1-\varepsilon_1}{\varepsilon_1} + \frac{1}{F_{12}} + \frac{A_1(1-\varepsilon_2)}{A_2 \varepsilon_2}} \quad (\text{S21})$$

where A and ε are the surface area and emissivity respectively, σ is the Stefan–Boltzmann constant. Subscripts 1 and 2 means the objects 1 and 2. In our case of a very small nanoribbon (object 1) surrounded by a much larger enclosure (object 2), $F_{12} = 1$, and $\frac{A_1}{A_2} \cong 0$. The emissivity of nanoribbon ε_1 is very small. Therefore, the above equation is reduced to:

$$\dot{Q}_{12} = \varepsilon_1 A_1 \sigma (T_1^4 - T_2^4) \quad (\text{S22})$$

We can see that the radiative flux only depends on the emissivity of object 1 (i.e., nanoribbon in our case) and is independent of the properties of the enclosure (either the vacuum chamber wall or the Si substrate).

As mentioned earlier, $T_2 = T$ (enclosure or environmental temperature), $T_1 = T + \Delta T$ where $\Delta T \ll T$ (Note: ΔT is between ΔT_h and ΔT_s), so the radiative heat transfer coefficient (h_r) can be written as:

$$h_r = \frac{\varepsilon_1 A_1 \sigma (T_1^4 - T_2^4)}{T_1 - T_2} \cong 4\varepsilon_1 \sigma T^3 \quad (\text{S23})$$

Where ε_1 is the NR emissivity, which is obtained from the simulation described in Note 11 for different NR thicknesses. The radiation heat loss from the NR is calculated based on a thermal fin model. The steady state heat conduction equation along the NR can be written as:

$$kA \frac{d^2 \Delta T}{dx^2} - h_r P \Delta T = 0 \quad (\text{S24})$$

where k is the NR thermal conductivity, A is the cross-section area of the NR and P is the perimeter of the NR cross section.”

4. When deriving the value of emissivity, the author assumes infinitely long nanoribbons. Is there any dependency of emissivity on the length of nanoribbons? What do you mean by ‘long propagating modes’ in page 20 of Supplementary Note.

Response:

The emissivity is independent on the length of the nanoribbon for the length range we studied here. The length dependence of SiO₂ nanoribbons was studied by Carlos Cuevas group [Ref: García-Esteban, J. J., Bravo-Abad, J., & Cuevas, J. C. (2022). *ACS Photonics*, 9(11), 3679-3684.]. Their results (particularly Fig. 2 in their paper, also included below) show that as the length of the nanoribbon increases, the emissivity saturates at length of ~50 μm. The lengths of nanoribbons in our experiments are over 44 μm, so they can be well represented by infinitely long ribbons considered in our emissivity modeling.

Figure R2: Figure 2 from García-Esteban et al., *ACS Photonics*, 9(11), 3679 (2022), showing the (a) length and (b) width dependence of thermal emissivity of SiO₂ nanoribbons. Here, W , τ , L are width, thickness, and length, respectively, of the nanoribbons.

For the ‘long propagation modes’ we mean the propagation length of the SPhP modes supported by the nanoribbon is much longer than the actual length in the experiment. These long propagating modes are typically having large mode size which may lead to energy leakage to the Si substrate. This energy loss term was not accounted for in our emissivity model, but would be captured in our experimentally determined “emissivity” which measures the length dependent energy loss of nanoribbons of different lengths. This may explain the discrepancy between the modeled and measured emissivity shown in Figure S15: the discrepancy is bigger at lower temperature because the dominant wavelength is longer and the mode size (and hence the leakage) is larger.

We have modified **Page 22 of Supplementary Note 11** as follows for clarifying.

“At lower temperature, the simulation result underestimates the emissivity value. This is likely due to the more leakage of the SPhP waves to the substrate at lower temperatures. At lower temperatures, the peak thermal wavelength is longer and therefore the corresponding mode sizes of the propagating modes are generally larger, leading to more leakage. This part of the energy loss is included in the experiments but is not accounted for in the simulation, since it is very challenging to handle such a large simulation domain.”

5. When calculating Bose-Einstein distribution function, which value did you use as reference temperature? Is it mean temperature between hot and cold reservoirs?

Response:

We used the environmental temperature (T) as the reference for theoretical calculations. The temperature T is only slightly lower than that of the hot and cold reservoirs, as discussed in the response to question 3 above. In experiments, the temperature of the hot reservoir is $T_h = T + \Delta T_h$ and that of the cold reservoir is $T_c = T + \Delta T_c$. ΔT_h is about 2-30 K and ΔT_c is about 0.1-2 K. So, the average temperature of the hot and cold reservoirs is less than 16 K (or $\sim 5\%$) higher than the environmental temperature (which varies from 300-550 K). Therefore, it is a fair assumption to use the environmental temperature when calculating Bose-Einstein distribution function.

We have modified the manuscript as follows on **page 7 in the main manuscript**.

“ $f_\omega(T)$ is the Bose-Einstein distribution function and T is the environmental temperature”

6. When plotting Fig. 3b, what is the reference wavelength?

Response:

The reference wavelength is $10\ \mu\text{m}$. The previous Fig. 3b is a schematic figure, we now update it to a new plot based on real calculation data. It is the electric field intensity distribution (E^2) of the quasi-TM mode at the wavelength of $10\ \mu\text{m}$.

We have modified **Figure 3b in the manuscript** as follows.

Figure R3 (Figure 3b in the revised manuscript), Simulated mode profile of the SiO₂ NR waveguide quasi-TM mode at the wavelength of 10 μm in the thickness direction for two samples: $t = 22$ nm and $W = 0.93$ μm (blue) and $t = 44$ nm and $W = 9.9$ μm (black). The absolute value of electric field intensity for blue line is much smaller than black one. The intensity is normalized with their maximum value in the figure for better comparison. The thinner and narrower the NR, the larger the mode size, leading to SPhP energy leakage to the substrate.

7. For Fig. 3c, the authors argue that 1-μm-wide sample follows the T dependence. However, because it has very small cross section, the mode size should be very large, leading to significant leakage to substrate. Is it still reasonable that it is 1D heat conduction, which follows the T dependence, although there should be a large deviation between theoretical prediction and experimental result?

Response:

This is an excellent point. The reviewer is correct that the deviation from the theoretical limit of 1D or 2D heat conduction due to various loss mechanisms could make the measured SPhP conductance (G_{SPhP}) not exactly following the 1D or 2D temperature dependence. In fact, now we have done more quantitative T dependence analysis of our data and found that G_{SPhP} of the 1 and 10 μm wide samples have temperature dependence close to but not exactly equal to T and T^2 , respectively, as shown in Figure R4 below. The near 1D and 2D temperature dependence respectively for the narrow and wide sample is reasonable, despite the deviation of G_{SPhP} from the theoretical limit. This is because the loss mechanisms causing this deviation are mostly dependent on the mode size (as the reviewer pointed out), which is only weakly dependent on temperature within a narrow temperature range.

As the reviewer also mentioned, a large mode size can lead to significant leakage to substrate (and other losses), thus low G_{SPhP} . The measured G_{SPhP} of the 1-μm-wide sample in Fig. 3c is only about 10% of the 1D theoretical limit. However, for a narrow temperature range the percentage of the measured G_{SPhP} relative to the theoretical value is almost the same, so the measured G_{SPhP} also show similar T dependence as the theoretical prediction in within a narrow temperature range.

We can consider different loss mechanisms of the surface wave propagating in a SiO₂ nanoribbon. The guided wave on the SiO₂ nanoribbon will eventually end up with three results: 1. absorbed by the reservoir, 2. scattered into free space, 3. leakage into the surrounding structures during the propagation. The first term is what we want for the SPhP thermal conduction. The second term is induced when the mode profile on the SiO₂ NR does not match the mode profile on the reservoirs well. The third term (leakage to the surrounding) happens when mode size is too large and overlaps with the Si substrate. Therefore, for the second and third terms (both represent losses), the mode size is important, as the reviewer pointed out.

For a sample with specific cross-section, the mode size does depend on the wavelength, i.e., the contribution to the SPhP thermal conduction is affected by temperature since the peak thermal wavelength of the emission spectrum changes with temperature. However, for a narrow temperature range such as from 400 to 550 K, the peak thermal wavelength does not change dramatically. In Equation S16 we show the mode size (MS) as:

$$MS_{NR}(\omega) \propto \frac{1}{\tanh\left(\frac{tW}{C(\omega)}\right)}$$

where C is a parameter related to frequency and material property. For SiO_2 , the material property varies dramatically within the Reststrahlen band (8-9.3 μm) [Ref: Ordonez-Miranda et al., Journal of Applied Physics 115, (2014)] but changes slowly with wavelength outside the Reststrahlen band. The supported modes on the SiO_2 nanoribbon are much broader than the Reststrahlen band, so that the average mode size does not change strongly with temperature especially at relatively higher temperatures. Since the mode size dictates the fraction of the SPhP energy captured by the cold thermal reservoir, we suggest that the ratio of the measured G_{SPhP} to the theoretical limit is also weakly dependent on temperature within a narrow temperature window.

We have modified the manuscript as follows on **page 14 of the manuscript**.

“We fit the measured G_{SPhP} as a power law function of temperature ($G_{SPhP} \sim T^\alpha$), and find that α is close to 1 and 2 for the $\sim 1 \mu\text{m}$ and $\sim 10 \mu\text{m}$ wide samples, respectively within the temperature range of 400-550 K. (**Fig. 3c**). This is consistent with the theoretical prediction of 1D and 2D G_{SPhP} in narrow and wide SPhP waveguide respectively^{9,31}. However, the exponents α deviate from 1 or 2. This is due to the fact that the observed G_{SPhP} is still only a fraction of the respective theoretical 1D and 2D limits. For example, G_{SPhP} of the $\sim 1 \mu\text{m}$ wide samples is about 10% of the theoretical 1D limit. The deviations from the theoretical limit are caused by the losses of SPhP waves, either leaked into the substrate during propagating or scattered into free space when coupling into the cold reservoir. Both loss mechanisms depend on the mode sizes of the waves: the larger the mode size, the bigger the losses. The mode size mainly depends on the cross section of the sample and it is larger for samples with smaller cross sections. That is the reason for the lower G_{SPhP} in the $\sim 1 \mu\text{m}$ wide samples. For a specific sample, the mode size could also change with the temperature. This is mainly because the mode size depends on the wavelength, and the peak thermal wavelength of the emission spectrum depends on the temperature. However, for a narrow temperature range such as from 400 to 550 K, the peak thermal wavelength does not change dramatically. Furthermore, the spectral dependence of the mode size is also quite weak for the case considered here. For SiO_2 , the material property varies dramatically within the Reststrahlen band but changes slowly with wavelength outside the Reststrahlen band¹⁸, so does the mode size as shown in Supplementary Note 9. The modes on the SiO_2 NRs are supported with frequency range much broader than the Reststrahlen band, so the average mode size does not change strongly within a narrow temperature range especially at relatively high temperatures. That means the percentage of measured G_{SPhP} relative to the theoretical value is almost the same, so the measured G_{SPhP} also show similar T dependence as the theoretical prediction within the narrow temperature range studied.”

Figure R4 (Figure 3c in the revised manuscript). Temperature-dependent G_{SPH} of the SiO₂ NRs with different thickness and different widths of ~ 1 and ~ 10 μm , showing close to T and T^2 dependence respectively. The apparent thermal conductivity of these samples is shown in Fig. 2c.

8. In this manuscript, there are many factors which can affect both theoretical predictions and experimentally derived value of SPhP-mediated thermal conductivity. For example, when obtaining absorption efficiency, C is related to frequency and material property. The experimental data in Fig. 3d are obtained with nanoribbons with different cross section. Their mode size should be a function of wavelength and those nanoribbons should have different cut-off wavelength (i.e., wavelength wo leakage) because the gap between nanoribbons and Si substrate is 70 μm . However, the experimental data with different cross-sectional area are all fitted to single fitting parameter $C = 0.45 \mu\text{m}^2$. How is it possible and what is the physical meaning of this constant fitting parameter? If you could provide more detailed explanation, it would help general readers' understanding.

Response:

The parameter C is the characteristic cross-sectional area to be scaled with the physical cross-sectional area ($A = Wt$, W is the width and t is the thickness). We used a simple correlation of $\eta_n(\omega) \approx \tanh(A/C(\omega))$ (derivation in Supplementary Note 9) to fit the coupling efficiency for the data for our 1D samples (red data points in Fig.3d). To avoid confusion with the frequency dependency $C(\omega)$ in the $\eta_n(\omega)$ equation, we now use C_{eff} as the fitting parameter, which can be treated as an average scaling factor of the mode size for the four samples under the temperature of 550K.

The reviewer is correct that the parameter C could be affected by the cross section due to the change in the cut-off frequency (for the same ribbon-substrate distance of 70 μm). However, as explained in the

response to the previous comment (question #7), the spectral dependence of the mode size is weak outside the Reststrahlen band. For SiO₂, the material property could vary dramatically within the Reststrahlen band but changes slowly with wavelength outside the Reststrahlen band, so does the mode size as shown in Supplementary Note 9. The modes on the SiO₂ NRs are supported with a frequency range much broader than the Reststrahlen band (response to question #1), so the average mode size does not change strongly with the wavelength, especially at a high temperature. The data in Fig. 3d is for T = 550 K and the corresponding thermal emission peak is about 5.27 μm.

We have modified the Supplementary Note 9 as follows on Page 18.

“

$$\eta_n(\omega) \approx \tanh(tW/C(\omega)) \tag{S17}$$

The parameter $C(\omega)$ originated from the optical material property, which is frequency dependent. However, the spectral dependence of the mode size is weak outside the Reststrahlen band. For SiO₂, the material property could vary dramatically within the Reststrahlen band but changes slowly with wavelength outside the Reststrahlen band, so does the mode size as shown in Supplementary Note 9. The modes on the SiO₂ NRs are supported with a frequency range much broader than the Reststrahlen band, so the average mode size does not change strongly with the wavelength, especially at a high temperature. For example, at the temperature of 550 K, the corresponding thermal emission peak is about 5.27 μm which is already away from the Reststrahlen band.

We simply used $\eta_n = \tanh(A/C_{eff})$ to fit the data for our 1D samples (red data points in Fig.3d), where $A = tW$ is the cross section and C is an effective parameter after considering the frequency dependence. The trend aligns well when a single value of $C_{eff} = 0.45 \mu\text{m}^2$ is used (Fig. 3d in the main text). The fitted value of C_{eff} for our experiment data can be treated as an average scaling factor of the mode size for the four samples under the temperature of 550K.”

We also modified **Fig. 3d** to clarify the 1D fitting corresponding to the four small cross-sections (1D-like) samples.

Figure R5 (Figure 3d in the revised manuscript) Dependence of G_{SPhP} on the NR cross-section area compared to the calculated phonon thermal conductance of 50- μm -long NRs (solid black line) and the 1D and 2D quantum limits of G_{SPhP} (red and green dashed lines) in the small and large cross-section area regimes respectively. The experiment data corresponding to the $\sim 50\text{-}\mu\text{m}$ -long NRs in **Fig3. a**. Red dots for samples with small cross section (1D), green dot for sample with larger width (2D). The blue solid line is the fitting of the experimental data (red dots) to the 1D theoretical G_{SPhP} using Eq. (1). All the data shown in **d** are for 550 K.

Reviewer #2 (Remarks to the Author):

The authors investigated the enhanced thermal conductivity due to surface phonon polaritons (SPhPs) in SiO₂ nanoribbon waveguides. They measured thermal conductivities of different samples with different structural parameters and with/without Fe₃O₄ absorber. The comparison clearly shows the contribution of SPhPs to the thermal conductivity. The authors carefully design the structure and examine the data to demonstrate SPhP thermal conduction. The level of sample fabrication and measurement is very high. The manuscript is well written and contains useful information for the heat transfer community. However, some of the authors' claims need careful consideration and there are several data deficiencies. This manuscript may be reviewed and considered for publication after the revision.

Response:

We appreciate the reviewer's recommendation about our work and the constructive comments that have improved the manuscript in the revised form.

1. In p3, "Our results are the first time to demonstrate an enhanced effective thermal conductivity due to SPhP..."

Ref. 13 reported an enhanced thermal conductivity due to SPhP. Therefore, this claim is incorrect. The authors need to limit the novelty to the correct range and describe it accurately without overextending it. Also this claim should be updated throughout the manuscript.

Response:

We have removed the wording of "first time" in the text. Instead, we wrote "Our results **unambiguously** ~~are the first time to demonstrate~~ an enhanced effective thermal conductivity due to SPhP" on page 3.

2. In the abstract, the authors wrote "direct observation". What does direct mean? The authors compare the thermal conductivity with and without an absorber, which can be called indirect. Ref. 13 compares the thermal conductivities at different temperatures to prove that SPhPs are heat carriers. I think both are at similar levels. Also, "direct" appears on the bottom line of p. 8.

Response:

By "direct" we meant to say the measured thermal conductivity is higher than the well-known phonon limit in SiO₂, so the extra thermal conductivity can be attributed to the SPhP contribution. However, we agree that the word "direct" has no clear definition and could lead to different interpretations by different readers. Therefore, we removed the "direct" and "first direct (observation)" on p.8. Here is a list of these changes:

In Abstract, "Here, we realize direct observation of thermal conductivity mediated by SPhP in SiO₂ nanoribbon waveguides..." is changed to "Here, we realize **direct clear** observation of thermal conductivity mediated by SPhP in SiO₂ nanoribbon waveguides..."

On p. 3, removed the word "directly" in "In this work, ..., we **directly** observe an increased thermal conductivity in nanoribbons (NRs) of SiO₂,"

On p. 8, "To our knowledge, this is the **first direct** observation of enhanced thermal conductivity contributed by SPhP." is changed to "**This work provides a clear experimental observation of enhanced thermal conductivity contributed by SPhP through rational design of absorbers.**"

3. Why are there no error bars in Figure 2a, although there are in other figures? Please also clarify how the error bars are determined in the manuscript.

Response:

Thank you for pointing this out. We have added the error bars in Figure 2a as follows.

Error bars in Figure 2 are determined by the standard deviations from the conductance measurements and NR's dimension determination, following previously published protocols [Nano letters 11, 5507-5513 (2011); Nature communications 10.1 (2019): 1377]. Conductance measurement error originates from the linear regression fitting applied to the temperature rise versus power plots (Fig. S6). NR dimension determination error arises due to non-uniform width of NRs.

We have modified Figure 2a and caption in the manuscript as follows.

Figure R6 (Figure 2a in the revised manuscript). a, Comparison of thermal conductivity of samples with and without the Fe₃O₄ absorber. Error bars in all figures are determined by the standard deviations from the conductance measurements and NR's dimension determination, following previously published protocols.^{24,25} Conductance measurement error originates from the linear regression fitting applied to the temperature rise versus power plots (Fig. S6). NR dimension determination error arises due to non-uniform width of NRs."

4. The authors schematically draw the mode profile of the waveguide modes in membranes in Fig. 3b. The authors should calculate and show the real mode profile to claim that the significant energy leakage to the substrate in a thinner membrane.

Response:

Thanks for the suggestion. We now replaced it with a new figure based on real mode profile of QTM mode at the wavelength of 10 μm . The conclusion about the significant energy leakage is the same.

We have modified Figure 3b in the manuscript as follows.

Figure R7 (Figure 3b in the revised manuscript). Simulated mode profile of the SiO₂ NR waveguide quasi-TM mode at the wavelength of 10 μm in the thickness direction for two samples: $t = 22 \text{ nm}$ and $W = 0.93 \mu\text{m}$ (blue) and $t = 44 \text{ nm}$ and $W = 9.9 \mu\text{m}$ (black). The absolute value of electric field intensity for blue line is much smaller than black one. The intensity is normalized with their maximum value in the figure for better comparison. The thinner and narrower the NR, the larger the mode size, leading to SPhP energy leakage to the substrate.”

5. If there is more leakage to the substrate in a thinner membrane, why is the thermal conductivity for $t = 16$ nm (green) higher than that for $t = 50$ nm (orange) in Fig. 2c?

Response:

The SPhP contributed thermal conductance of 16 nm thick sample (~ 0.5 nW/K at 550K) is lower than 50 nm one (~ 0.8 nW/K) due to the larger loss (or leakage), but the thermal conductivity is higher due to the normalization to their cross sections: $k = \frac{G \cdot L}{W \cdot t}$. The two samples have similar width and length, but the 16 nm one has much smaller thickness, so the thermal conductivity is higher.

We have modified the manuscript as follows on page 12.

“It can be seen from **Figure 3c** that the SPhP contributed thermal conductance of 16 nm thick sample (~ 0.5 nW/K at 550K) is lower than 50 nm one (~ 0.8 nW/K) due to the large loss. The two samples have similar width and length, but the 16 nm one has much smaller thickness, its thermal conductivity is higher (**Figure 2c**) due to the normalization to their cross sections: $k = \frac{G \cdot L}{W \cdot t}$.”

6. In Fig. 3b the authors draw lines of T^1 and T^2 for the small W and large W structures respectively. It misleads the analysis and is not as convincing to conclude that these are 1D and 2D Gsphp. The authors should fit the experimental data and obtain the value of α (T^α) and discuss whether these are close to 1 or 2. Then please conclude that these are 1D or 2D Gsphp.

Response:

This is a great suggestion. The figure below shows the fitted value of α (T^α) of the experimental data from 400 to 550K. For ~ 1 μm wide samples, the fitted α is 0.96 and 0.72 for 49 μm and 99 μm long samples, respectively, which is close to the T linear dependence for 1D theoretical prediction. For ~ 10 μm wide samples, the fitted alpha ranges from 2.06 to 2.47 for different samples, which is close to T^2 dependence predicted by the theory for a 2D waveguide. Note that the fitting results are not exactly equal to 1 or 2. As explained in our response to question 7 of reviewer 1, this is due to the fact that the observed G_{SPhP} is still only a fraction of the respective theoretical 1D and 2D limits. For example, G_{SPhP} of the ~ 1 μm wide samples is about 10% of the theoretical 1D limit. The deviations from the theoretical limit are caused by the losses of SPhP waves, either leaking into the substrate during propagating or scattering into free space when coupling into the cold reservoir. Both loss mechanisms depend on the mode sizes of the waves: the larger the mode size, the bigger the losses. The mode size mainly depends on the cross section of the sample: it is larger for samples with smaller cross sections. That is the reason for the lower G_{SPhP} in the ~ 1 μm wide samples. For a specific sample, the mode size could also change with the temperature. This is mainly because the mode size depends on the wavelength, and the peak thermal wavelength of the emission spectrum depends on the temperature. However, for a narrow temperature range such as from 400 to 550 K, the peak thermal wavelength does not change dramatically. Furthermore, the spectral dependence of the mode size is also quite weak for the case considered here. For SiO_2 , the optical property varies dramatically within the Reststrahlen band but changes slowly with wavelength outside the Reststrahlen band [Ref: Ordonez-Miranda et al., Journal of Applied Physics 115, (2014)], so does the mode size as shown in Supplementary Note 9. The modes on the SiO_2 NRs are supported with frequency range much broader than the Reststrahlen band (response to

question 1, reviewer 1), so the average mode size does not change a lot with temperature especially at relatively high temperatures.

Since the measured G_{SPhP} are lower than the respective 1D and 2D theoretical limits, it's difficult to conclude these are 1D or 2D G_{SPhP} . Our experiment data only show temperature dependence similar to that of the theoretical 1D or 2D G_{SPhP} for narrower and wider samples, respectively. We have acknowledged this point in the Discussion section, but now with added discussions. In the future, if the total absorption efficiency can be improved to near perfect, one can then use SPhP to observe 1D and 2D quantum thermal conductance

Figure R9 (Figure 3c in the revised manuscript). Temperature-dependent G_{SPhP} of the SiO₂ NRs with different thickness and different widths of ~ 1 and ~ 10 μm , showing close to T and T^2 dependence respectively. The apparent thermal conductivity of these samples is shown in Fig. 2c.

We have modified Fig. 3c and the manuscript as follows on **page 14 and 16**, separately.

On page 14, added “We fit the measured G_{SPhP} as a power law function of temperature ($G_{SPhP} \sim T^\alpha$), and find that α is close to 1 and 2 for the $\sim 1 \mu\text{m}$ and $\sim 10 \mu\text{m}$ wide samples, respectively within the temperature range of 400-550 K. (Fig. 3c). This is consistent with the theoretical prediction of 1D and 2D G_{SPhP} in narrow and wide SPhP waveguide respectively^{9,31}. However, the exponents α deviate from 1 or 2. This is because the observed G_{SPhP} is still only a fraction of the respective theoretical 1D and 2D limits. For example, G_{SPhP} of the $\sim 1 \mu\text{m}$ wide samples is about 10% of the theoretical 1D limit. The deviations from the theoretical limit are caused by the losses of SPhP waves, either leaked into the substrate during propagating or scattered into free space when coupling into the cold reservoir. Both loss mechanisms depend on the mode sizes of the waves: the larger the mode size, the bigger the losses. The mode size mainly depends on the cross section of the sample, and it is larger for samples with smaller cross sections. That is the reason for the lower G_{SPhP} in the $\sim 1 \mu\text{m}$ wide samples. For a specific sample, the mode size could also change with the temperature. This is mainly because the mode size depends on the wavelength, and the peak thermal wavelength of the emission spectrum depends on the temperature. However, for a narrow temperature range such as from 400 to 550 K, the peak thermal wavelength does not change dramatically. Furthermore, the spectral dependence of the mode size is also quite weak for the case considered here. For SiO_2 , the material property varies dramatically within the Reststrahlen band¹⁸, so does the mode size as shown in Supplementary Note 9. The modes on the SiO_2 NRs are supported with frequency range much broader than the Reststrahlen band, so the average mode size does not change strongly within a narrow temperature range especially at relatively high temperatures. That means the percentage of measured G_{SPhP} relative to the theoretical value is almost the same, so the measured G_{SPhP} also show similar T dependence as the theoretical prediction within the narrow temperature range studied.”

On page 16, “This result suggests that one could realize the quantum limit of thermal conductance at room and high temperatures if the absorption coefficient can be enhanced to near unity, for instance, by using more optimized absorber materials and geometry .”

REVIEWERS' COMMENTS

Reviewer #1 (Remarks to the Author):

In the revised version of the manuscript, the authors have properly addressed my previous comments and I recommend publication.

Reviewer #2 (Remarks to the Author):

The authors have provided satisfactory answers to all questions and have made appropriate revisions. The revised manuscript has scholarly value worthy of publication in this journal and is recommended for publication.